# Factors affecting repurchase intention of organic food among generation Z (Evidence from developing economy)

**Muhammad Yaseen Bhutto[1], Mussadiq Ali Khan[2]\*, Chaojing Sun[3], Sharizal Hashim[4], Hassan Talal Khan[5]**

**1** Business School, Shandong Jianzhu University, Jinan, China, **2** Lahore Garrison University Lahore, Lahore, Pakistan, **3** Shandong Labor Vocational and Technical College, Jinan, China, **4** Faculty of economics and Management, Universiti Kebangsaan Malaysia, Malaysia, **5** Department of economics and management, Xi'an University of Technology, Xi'an, China

\* khanconnexion@yahoo.com

**Data Availability Statement:** The data set used during the current study can not be shared due to respondent's privacy concerns. As such, participants were assured that data will only be

## Abstract

Organic food has gained much importance due to consumers' rising environmental and health concerns. Purchase intention of organic food has been explored widely, but the repurchase intention of organic food has gained little attention among researchers. So, it has become important to explore repurchase intention among generation Z; a generation considered more educated and aware of rising environmental concerns. Generation Z is more tech-savvy and brand conscious, so its impact on repurchase intention through consumer satisfaction has been explored. The data in this paper was collected from 400 respondents through a structured questionnaire in Islamabad, Pakistan. We used the PLS-SEM approach for data analysis and results; we found that social media influence and brand purchase impact brand awareness and positively impact brand awareness on consumer satisfaction. Moreover, it is also found that consumer satisfaction positively impacts the repurchase intention of organic food. Our study found that Generation Z has a strong social media influence, so marketers' managers must consider and address the issues when consumers consider social media for their concerns and suggestion.

## 1. Introduction

Human health and environmental impacts are linked, and this is the main concern for environmentalists and academia today [1, 2]. Rapid economic growth and rising industrialization are leading to many environmental problems like depletion of natural resources and health-related issues [3]. These negative impacts on human health and environmental hazards compel consumers to change their food preferences [4]. As a result, organic food consumption is rising on the fast track [5]. Organic food is produced through organic agriculture techniques and without opting for the traditional ways, including bioengineering or usage of fertilizer [6]. Customer intention to purchase sustainable products is a significant driver to consuming organic food [7]. Organic food is considered healthier, has better taste, andis fresher than traditional food products [8, 9]. Because of multiple advantages, there is a wide range of

used for research purpose and will not be shared to third party for any reason. Therefore, data can not be shared for publicly used. However, those who are interested in datasets can request specially to ethical committee of UNIMAS Malaysia, on her email arrossazana@unimas.my.

**Funding:** NO-the funders had no role in study design, data collection and analysis, decision to publish, or preparation of the manuscript.

**Competing interests:** The authors have declared that no competing interest exist.

increasing organic food sales worldwide. Due to its advantages and benefits, sixfoldsales has been noted. It has increased from $15 billion in 1999 to $90 billion in 2016, with the U.S. having the highest share with 45 $ billion while E.U. stood second with 33.5$ billion [10].

It is noted that various studies have been conducted in western countries, but very few studies investigated environmental behaviour and related aspects in Asian countries [11]. Moreover, researchers like [12] hint at green initiatives taken by hotel industry in developing countries while [13] explainthe green sustainability of SMEs in developing countries. Furthermore [14, 15], explains the role of sustainability and CSR in the Pakistani manufacturing industry. Similarly [16], also focused on studying environmentally friendly consumption in developing countries. Moreover, it is found that 90% of organic food consumption is in developed countries while it is cultivated in developing countries like South Asian countries [17]. Pakistan also belongs to South Asia, and 20% GDP is rooted in agriculture. Out of 22.68 million hectares, 45299 hectares belong to organic, but it has only a 0.1 organic food industry worldwide [18, 19]. It is also known in research studies that Pakistanis spends half of their income on food (47%), while U.S. household spends a low share of income on food (6.6%) [20, 21]. It is also found that higher food-related deaths are recorded in Pakistan [16].

In Pakistan, organic food has been discussed from other aspects like [17] explaining the influence of different factors on the purchase intention of organic food. On the other hand [22], about the willingness to pay for organic food in Pakistan. Moreover [23], discussed applying the theory of planned behaviour topurchasing organic food. While [24] investigated the purchase intention of organic food in the Pakistani market. It is noted that researchers are not only limited to purchase intention and purchase behaviour of organic food in developing economies like Pakistan and scarcely discussed the intention to purchase of organic food. These facts make it relevant to study more organic food and its repurchase intention in Pakistan. It is also noted that generational impact is understudied. Literature shows a difference between all generation segments [25] For instance, generation Y are considered more educated and able to conduct multiple challenging tasks [26] (Chaturvedi, Kulshreshtha, & Tripathi, 2020). Regarding different generations and organic food, generation X and generation Y have been discussed [27] and narrated about the awareness of organic food among generation X and generation Y. Moreover [28], investigated about purchase intention of organic personal care products among generation X and Y. In addition to it, investigated antecedents of organic food purchases among generation Y [29]. Other researchers [30–41] discussed generation Y purchase behaviour of organic food. It has become evident that generation Y and generation X are focused, but generation Y and repurchase intention are rarely focused.Following emerging consumer generation is generation Z, considered more educated and well aware ofthe environment and environment-friendly products [42]. Generation Z is health conscious andpays attention to food and its nutrient value [43]. They are get influenced by social media influence [44]. Young Pakistani customers are tilted to purchase more organic products [45]. Witnessing the Pakistani scenario, the young generation is nearly 65% of the total population [19]. While this study presents a comprehensive picture of generation Z, it is found that generation Z is notoriously understudied.So it is pertinent to study them and make it relevant to deeply understand generation Z repurchase phenomenon.

In recent times, communication has changed the whole way of interaction and influence, and social media has become its recent example. The rise of social media enables consumers to collect and disseminate information about their consumers. For example, different social media websites, Vlogs, written blogs and different social media channels made the consumers and stakeholders engage with each other's irrespective of their geographical presence [46]. It is noted that social media shape attitude towards organic food [47]. Adding to this, it is also concluded that social media has a significant role on consumers' attention, interest, and search

[48]. Social media influence is obvious in creating awareness among generation Z as this generation group is much more involved and comfortable using new technologies.

Secondly, the role of purchase is also important in consumer awareness. Without interaction and experience with the desired product, it is almost impossible for consumers to understand the product. Consumers' decisions are taken with utmost care and consist of many phases, so it contains a lot of steps. So, the brand purchase has the utmost significance in creating brand awareness.Consumer satisfaction and repurchase intention are important concepts to explore, and it has to see how they are interrelated. Customer satisfaction positively influences customer loyalty [49], and loyal customers are willing to purchase many times.It has been noted that the purchase intention of organic food has been largely studied in different contexts [50]. But it is noted that the repurchase intention of organic food is scarcely investigated, and few studies have been conducted [51, 52], studies on repurchase intention of organic food in which they probed the impact of environmental awareness, healthy consumption, consumer attitude and perceived price fairness, perceived quality was investigated. Furthermore [53], have studied the repurchase intention of organic food in India and the impact of price fairness, perceived consumer social responsibility, perceived value, and perceived quality on the repurchase intention of organic food studied. Still, no study of the repurchase intention of organic food has been studied in Pakistan.

It is concluded that social media, brand awareness, and the role of purchase are significant in investigating consumer satisfaction and repurchase intention of organic food. It would be interesting to understand the impact of the role of all these factors and how these factors are linked to the repurchase intention of organic food.

Generationcohort is termed as different age groups who are born at different periods at some specific place and have gone through different experiences during their coming age like 17–23. So, generational cohorts are considered effective in mapping out consumer marketing, known as new segmentation. So, generational cohort theory effectively describes this phenomenon of different age groups like generation X, Y, & Z. Generation Cohort theory application has been fairly limited and has not been implemented to repurchase intention of organic food and generation Z.Different generation groups like generation X and generation Y are part of Pakistani population. Still, Generation Z is one of the biggest age groups inthe Pakistani consumer market. The impact of generation Z and its link with repurchase intention is a new phenomenon to explore and would be quite a novel study. So, it would be significant to understand how generation Z behaves and purchases organic food products and how it is important to investigate and pivotal for repurchase purchase intention.

Pakistani organic market has a lot of potential to explore, so this study can explore new avenues of research.Generation Z can attract companies to further invest in this area and target generation Z in bringing up their products, and newness of this study is multi-fold.

## 2. Literature review

It is noted that currently, the organic food segment is making a remarkable effort to execute various social and environmental measures and processes, and the impact of nutritional products that are considered significant and critical for healthy food is adding value to consumers' diets [54]. In given choices, consumers have limited ability to process the information. At the same time, when choices are available before consumers then surely on given choice criterion, consumers process up some information, evaluate different options, and eliminate various alternatives through comparison to measure different sets of products and brands available [52, 55, 56]. Regarding food, healthy behaviour is affected by certain barriers like limited access to food, eating habits, cost, time and intention to consume [57]. Consumers are more interested

in organic food for multiple reasons; they perceive that organic food is better for health, environmentally friendly, produced through traditional cooking, brings a healthier diet, assists the local economy and has more advantages for health than inorganic food [52]. It is important to note that organic food has potential benefits for health, and it is produced through environmentally friendly methods, including no use of any chemicals or pesticides or any genetically modified organism, chemical additives or industrial solvents [58]. Generation cohort theory proposes that individuals belonging to the same generation will have the same economic, political and social events during their early phase of life and will develop the same set of beliefs, values and behaviour [59–61]. It hints that events shape that particular generational cohort in their particular birth year and after 10 to 20 years [61]. This theory paves the best way to analyse market segmentation, which is the fine approach to consumer analysis [62, 63]. An organization must learn to utilize the latest platform like social media to attract youth, especially generation Z, which is well versed with these kinds of forums. However, it is significant to inform employees and consumers ofthe environmental advantages of sustainable products. Moreover, realizing how the environment can impact consumers' daily lives and their decision power will help them train themselves and better understand environmentally friendly products [63, 64].

## 2.1 Social media influence

Generation Z affiliates are well-educated consumers who better understand sustainability issues and sustainable products [1]. Being educated and well-versed with the latest platform, social media has become part and parcel of the daily lives of youth, especially generation Z. Various forms of social media like Facebook, Twitter, YouTube, WhatsApp, Instagram etc., help to create personalized online pages and communication and interaction with friends and exchange context which they have created their own and some information which is related to other brand-related sources [65, 66]. New media and Technology are part of the daily life of young adults [67, 68]. Many factors are significant for adopting healthy behaviour, and social media is also important to know about healthy behaviour [69]. It is noted that young adults spend most of their time with Technology, so their influence on social media is evident [70]. So, the study can hypothesize that

H1: Social Media has a positive impact on brand awareness.

## 2.2 Brand purchase

Consumers decide onthe purchase after various stages of evaluation. So, brand purchase is important for both consumers and brands. If the brand purchase is taken with considerable thought, consumers feel experienced after every purchase decision. Making consumers much more aware is the essence of the brand purchase decision. Artificial intelligence technologies help the purchase of retailing products [71]. Artificial intelligence can accurately predict brand purchase patterns [72]. In a nutshell, brand purchase is more critical for brand awareness because, through a series of evaluations and close interactions with the brand, consumers feel more aware and experienced. So, the study hypothesizes as

H2: Brand purchase has a positive impact on brand awareness.

## 2.3 Brand awareness

It is significant to note that better consumer awareness about the environment and quality of life concerns will lead to the better pursuit of organic food as these are the main pillars of sustainable, healthy and refined choice of a healthy diet [51]. It is also known that healthy food

pursuit is the main reason consumers prefer organic food [51]. So, it has become obvious that consumer awareness is needed to choose organic food.

Brand awareness results from many experiences, including the purchase of that product. When a brand is chosen for purchase, consumer is more aware of that particular brand for the next time selection or preference of that brand. Brand awareness impact on buying Islamic insurance [73]. Brand awareness can provide the reason forthe existence, core and commitment that is significant for the company [74]. In addition, social media's influence is vital for consumers' brand awareness [75]. Awareness positively impacts behaviour [76]. So, the study hypothesizes as

H3: Brand awareness has a positive impact on consumer satisfaction

## 2.4 Consumer satisfaction

Consumer satisfaction is defined as a feeling of pleasure of a person or disappointment that results from comparingthe performance and quality of a product against the expected product [77]. In today's competitive environment, brands are more tilted towards providing better value to theirconsumers so that satisfied consumers will be their assets and will not be easily lured by competitors. So, the satisfaction of consumers is considered important for companies. In the service industry, customer satisfaction also plays a vital role [78]. The usage of the product by the consumer can only reveal consumer satisfaction [79]. It is deduced from consumer satisfaction that similar products will be assessed and used for consumption in the future. [74, 80] concluded that customer satisfaction plays pivotal role in the online environment exhaustively to hold the consumer journey. Product performance is reflected in perceived satisfaction, which makes it eligible to recommend the product to others [81]. Customer satisfaction positively influences loyalty and WOM [82]. So, it has become evident that repeated product purchases are made through the recommendation of other consumers. So, mediation of consumer satisfaction is suitable for the study.

H5: Consumer satisfaction positively impacts on repurchases intention

## 3. Materials and methods

The main focus of the study was Generation Z (people born between 1995–2012) [83]. This reflects that generation has information and approach towards product category which make it improve to conduct a study on their particular behavior. Moreover, the study was conducted towards educated consumers to receive better output. Furthermore, this generation already knows the concept of brand awareness, satisfaction and repeated purchase. Using quantitative methods, data collection was done through a survey approach. Data collection was made possible through a self-administrated questionnaire, and consumer responses were recorded. The questionnaire consisted of two parts which were in the English language. The first part comprised demographic information about gender, age, income and educational qualification. The second part measured repurchase intention, consumer satisfaction, and brand awareness.

### 3.1. Data collection and sampling techniques

The questionnaire was distributed among organic food consumers in Rawalpindi/Islamabad (twin cities of Pakistan). Local students were hired for data entry. It was assured to consumers that the study is conducted for research and secrecy of the study would be maintained. Convenience sampling was used as it is better for collecting information when challenging, and its

**Table 1. Respondent profile.**

| Details | | Frequency | % |
|---|---|---|---|
| Gender | Male | 220 | 55 |
| | Female | 180 | 45 |
| Age (Years) | 18–30 | 195 | 48.75 |
| | 31–40 | 170 | 42.5 |
| | 41–50 | 32 | 8.00 |
| | Above 50 years | 3 | 0.75 |
| Educational qualification | Bachelors | 203 | 50.75 |
| | Masters | 122 | 30.5 |
| | M.Phil. | 50 | 12.5 |
| | Ph.D. | 25 | 6.25 |
| House Hold Yearly | Above or equal to 60000 RS | 155 | 38.75 |
| | Less than equal to 60000–100000 Rs | 145 | 36.25 |
| | More than 100,000 Rs | 100 | 25.00 |
| | | | |
| | | | |

output is suitable for generalization. For data collection among 400 organic food consumers and out of 400, 350 samples were received with a response rate of 87.5%.

## 3.2. Measurement and survey instrument

The research tools have been borrowed from previous research studies of a similar kind. The English language was used, and responses were received from "strongly disagree to agree strongly". Measurement of repurchase intention (R.I.) was comprised of five items and was adapted from the study of [84, 85]. Measurement of Consumer satisfaction (C.S.) consisted of four items adapted from [86]. Brand awareness measurement consisted of five items adapted from [87].

The respondents' demographic characteristics are as follows: 220(55%) males and 180 (45%) females were surveyed. All the respondents were adults; 195 (48.75%) were from the age group (18–30), and 170 (42.5%) were from 31–40, in that order followed by age group. Education-wise, 203(50.75%) had a Bachelor's degree, 122(30.5%) had master's level education, 50 (12.5%) had M.Phil., and the remaining 25(6.25%) had PhD level education. The income of a household was as follows, 155(38.75%) earn less than or equal to 254 USD yearly, 145 (36.25%) belongs to earning group 236–424 USD, and 100(25%) earn more than 424 USD yearly (see Table 1).

## 4. Results

### 4.1. Data analysis and results

The study used structural and analytical modelling methods to create a model for repurchase intention. The study used the following methods: Confirmatory factor analysis (CFA) and structural equation (SEM).

Convergent validity was evaluated on three main conditions (1) the standardized factor load exceeds values of 0.5 (2) composite reliability (C.R.) was higher than extracted mean deviation (AVE, average variance extracted) and (3) according to the standard suggested by Hair et al., [88] AVE exceeded 0.5. Furthermore, Table 2 shows all the values of C.R. and AVE are 0.846 to 0.901 and 0.579 to 0.752 by standard values of 0.7 and o.5, respectively [89].

**Table 2. Data analysis and results.**

| Variables | Items | Loading | Composite Reliability | Average Variance Extracted (AVE) |
|---|---|---|---|---|
| Consumer Satisfaction | | 0.807 | 0.886 | 0.722 |
| | CS1 | 0.858 | | |
| | CS2 | 0.844 | | |
| | CS3 | 0.847 | | |
| Brand Awareness | | 0.758 | 0.846 | 0.579 |
| | BA1 | 0.807 | | |
| | BA2 | 0.763 | | |
| | BA3 | 0.707 | | |
| | BA4 | 0.763 | | |
| Brand Purchase | | 0.764 | 0.863 | 0.678 |
| | PI | 0.783 | | |
| | PI2 | 0.864 | | |
| | PI3 | 0.822 | | |
| Repurchase Intention | | 0.835 | 0.901 | 0.752 |
| | RI1 | 0.886 | | |
| | RI2 | 0.854 | | |
| | RI3 | 0.861 | | |
| Social Media Influence | | 0.770 | 0.852 | 0.593 |
| | SMM1 | 0.644 | | |
| | SMM2 | 0.804 | | |
| | SMM3 | 0.861 | | |
| | SMM4 | 0.757 | | |

For discriminant validity, square root values of AVE should exceed its correlation coefficient with other constructs [90]. According to this model explanation, the researcher studying this model should compare the square root of (AVE) of each construct with the shared variance between the construct. If the AVE's square root is better than the shared variance between the constructs, the researcher can announce that the construct has discriminant validity. Table 3 shows that AVE's square root of each construct was higher than its correlation coefficient with other constructs, indicating that discriminant validity is achieved.

The hypothesis was tested through the usage of structural equation modelling (SEM), and the results are presented in Table 4, brand purchase (0.387, t value, 8.683, p values, 0.000), social media influence (Beta, 0.441,T-value 10.300,P-value,0.000) have a significant impact on brand awareness. While brand awareness (beta value 0.648,t value 13.714, p-value 0.000) indicates a positive impact of brand awareness on consumer satisfaction. Consumer satisfaction (Beta 0.660, T-15.408, P 0.000) shows that there is positive and significant relationship with repurchase intention.

**Table 3. Correlation matrix.**

| Variables | Consumer Satisfaction | Brand Awareness | Brand Purchase | Repurchase Intention | Social Media Influence |
|---|---|---|---|---|---|
| Consumer Satisfaction | 0.85 | | | | |
| Brand Awareness | 0.648 | 0.761 | | | |
| Brand Purchase | 0.37 | 0.626 | 0.824 | | |
| Repurchase Intention | 0.66 | 0.663 | 0.519 | 0.867 | |
| Social Media Influence | 0.528 | 0.651 | 0.544 | 0.553 | 0.77 |

**Table 4. Relationship.**

| Hypothesis | Beta | T-value | P-Value |
|---|---|---|---|
| Brand Purchase -> Brand Awareness | 0.387 | 8.683 | 0.000 |
| Social media influence -> Brand Awareness | 0.441 | 10.300 | 0.000 |
| Brand Awareness ->Consumer Satisfaction | 0.648 | 13.714 | 0.000 |
| Consumer Satisfaction -> Repurchase Intention | 0.660 | 15.408 | 0.000 |

## 5. Discussion

Knowing we established a hypothesis and in it, brand awareness is very important in assessing the brand's quality. In addition to it, brand awareness is critical for achieving excellence among consumers. So, there are other factors as well that are influence brand awareness, like brand purchases. When consumers buy a product, he has to go through different phases of decision-making and product usage phases. Product usage gives him much more knowledge about certain products, and the consumer is much more aware after consuming the product, unlike before. Generation Z is a relatively inexperienced age group, unlike generation x or baby boomers. Their approaches towards certain products are vague until they purchase the product, and they may get the experience through the purchase. It is also possible that generation Z is aware of them through more encountering with a certain product. So, post-purchase response and after consumption of the product put them in a position to consider they more compact and solid about a certain product. This shows that brand purchase has a positive impact on brand awareness and the result of the study confirms this relationship. Secondly, Our study found a positive relationship between social media marketing and brand awareness; the results support past studies [75, 91]. Social media is in much usage of generation Z. Hence, it is important to note the importance of social media marketing among generation Z. Social media marketing is significant for brand awareness. Hence, it is possible that generation Z got influenced by brand awareness of the organic product.

Our study found a positive relationship between brand awareness and consumer satisfaction, whichendorses past study [74]. It shows that when consumers are much more aware, their satisfaction is correlated. It is possible that when consumers get more awareness about certain products, their satisfaction improves because their knowledge improves, which enhances their trust in certain brands.

Our study found a positive relationship between consumer satisfaction and repurchase intention of organic food, findings parallel to past studies [92, 93]. It shows that consumer satisfaction improves trust in products, matches expectations and compels consumers to invest their money in purchasing the product for particular brands.

### 5.1. Conclusions (Theoretical and practical contributions)

**Theoretical contribution.** This study focused on the least focused area of organic food as repurchase intention of organic food. It would help to understand youth psychology and level of comprehension of organic food. Moreover, it would help new researchers focus on this point voraciously as a base point.

Secondly, it is noted that the impact of social media influence and brand purchase has been understudied, and this study instigated this discussion to make it a focal point. Generation Z is aggressively using social media, so the study pertinently provided an overview of this. The effect of brand purchase on the repurchase intention of organic food is a milestone in comprehending this phenomenon.

Thirdly, our contribution relates to unexplored generation Z, who experienced a lot and the unexplored age group in the organic food sector. This study mainly focused on how different

factors are important for the repurchase intention of organic food, and this study is a milestone in developing an understanding of generation Z. It is deduced from the study that social media has a critical influence on Generation Z, so brand managers should focus while making up their marketing and advertising plan for their organic food brand. It also shows the need for rigour in social media activism to retain a presence among generation Z consumers.

Fourthly, developing economies like Pakistan, which has the 6th largest population in the world, rarely focus on environmentally friendly products. This study attempted to fill this vacuum and started a debate about this topic, another study contribution.

**Practical contribution.** Our findings have significant theoretical and practical contributions.Brand awareness, customer satisfaction, brand purchase and social media influence are the factors which influence customers' purchase decisions. Moreover, marketing managers know the importance of these factors and always try to manage social media to influence brand awareness and customer satisfaction.The current study would assist marketing managers for several reasons.First, our study showed a positive impact of social media influence and brand purchase on brand awareness among generation Z consumers. Results confirm that social media influence and brand purchase positively impact brand awareness. Generation Z in Pakistan isthe wide majority in Pakistan (almost 65.5, according to the Pakistan Bureau of Statistics). So, it shows that social media influence must be considered while making up advertisements. Secondly, it is also important to note that due to social media influence, awareness among the masses for organic food is also rising. Thirdly, due to different social media platforms, it has also become easy to interact with different sellers for organic food purchases and repurchases. So, social media influence has multiple effects on the repurchase of organic food among generation Z. Results found that the influence of social media is important for brand awareness.

Secondly, the study also contributed to improving the understanding of the impact of brand awareness on consumer satisfaction—limited literature about brand awareness and consumer satisfaction among generation Z, particularly in organic food. So, the study contributed that there is an impact on brand awareness and brand satisfaction among generation Z.

Thirdly, government agencies and related departments must raise awareness among youth, particularly generation Z, about the benefits of organic food. There is a lack of solid policy in this regard. The government must incentivize farmers and activate other forums, including large agricultural research departments, to enhance their potential in this domain so that consumers can benefit from it.

**Future directions& limitations.** In this study, the focus was on the impact of brand awareness and consumer satisfaction on the repurchase intention of organic food among generation Z. This research was limited to the twin cities of Pakistan (Rawalpindi & Islamabad) and can be extended to other cultures. This study was limited to social media influence and can be extended to specific platforms like the role of Facebook pages in promoting and repurchasing organic food sectors. Future researchers focus on the impact of the brand price, brand promotion, and different aspects of social media influence on the repurchase intention of organic food. Future researchers can determine how these factors are important for the repurchase intention of organic food consumers. Moreover, this research was focused on generation Z of Pakistani organic consumers, but this research can be extended to generation Y of organic food consumers.

## Supporting information

**S1 Appendix.**
(DOCX)

## Author Contributions

**Conceptualization:** Muhammad Yaseen Bhutto, Mussadiq Ali Khan.

**Formal analysis:** Muhammad Yaseen Bhutto.

**Investigation:** Muhammad Yaseen Bhutto, Mussadiq Ali Khan.

**Methodology:** Mussadiq Ali Khan, Chaojing Sun.

**Project administration:** Chaojing Sun.

**Resources:** Chaojing Sun.

**Supervision:** Mussadiq Ali Khan.

**Writing – original draft:** Muhammad Yaseen Bhutto.

**Writing – review & editing:** Sharizal Hashim, Hassan Talal Khan.

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
