## [Decision Letter · Decision Letter 0]

8 Sep 2022

PONE-D-22-18927factors affecting repurchase intention of organic food among generation Z (Evidence from developing economy)PLOS ONE

Dear Dr. Khan,

Thank you for submitting your manuscript to PLOS ONE. After careful consideration, we feel that it has merit but does not fully meet PLOS ONE’s publication criteria as it currently stands. Therefore, we invite you to submit a revised version of the manuscript that addresses the points raised during the review process.

We look forward to receiving your revised manuscript.

Kind regards,

Ming Zhang, Ph.D.

Academic Editor

PLOS ONE

Journal Requirements:

3. You indicated that ethical approval was not necessary for your study. We understand that the framework for ethical oversight requirements for studies of this type may differ depending on the setting and we would appreciate some further clarification regarding your research. Could you please provide further details on why your study is exempt from the need for approval and confirmation from your institutional review board or research ethics committee (e.g., in the form of a letter or email correspondence) that ethics review was not necessary for this study? Please include a copy of the correspondence as an ""Other"" file.

4. We suggest you thoroughly copyedit your manuscript for language usage, spelling, and grammar. If you do not know anyone who can help you do this, you may wish to consider employing a professional scientific editing service. 

Reviewers' comments:

Reviewer's Responses to Questions

**Comments to the Author**

1. Is the manuscript technically sound, and do the data support the conclusions?

Reviewer #1: Yes

Reviewer #2: Yes

2. Has the statistical analysis been performed appropriately and rigorously? 

Reviewer #1: Yes

Reviewer #2: Yes

3. Have the authors made all data underlying the findings in their manuscript fully available?

Reviewer #1: Yes

Reviewer #2: Yes

4. Is the manuscript presented in an intelligible fashion and written in standard English?

Reviewer #1: Yes

Reviewer #2: Yes

5. Review Comments to the Author

Reviewer #1: The paper is quite relevant and interesting, but some improvements are necessary to make it even better.

The relationship between consumer perceptions and sustainable consumption behavior as regards repurchase intention of organic food has not been covered, and thus such sources can be cited:

Kliestik, T., Zvarikova, K., and Lăzăroiu, G. (2022). “Data-driven Machine Learning and Neural Network Algorithms in the Retailing Environment: Consumer Engagement, Experience, and Purchase Behaviors,” Economics, Management, and Financial Markets 17(1): 57–69. doi: 10.22381/emfm17120224.

Hopkins, E. (2022). “Machine Learning Tools, Algorithms, and Techniques in Retail Business Operations: Consumer Perceptions, Expectations, and Habits,” Journal of Self-Governance and Management Economics 10(1): 43–55. doi: 10.22381/jsme10120223.

Nica, E., Sabie, O.-M., Mascu, S., and Luțan (Petre), A. G. (2022). “Artificial Intelligence Decision-Making in Shopping Patterns: Consumer Values, Cognition, and Attitudes,” Economics, Management, and Financial Markets 17(1): 31–43. doi: 10.22381/emfm17120222.

Lăzăroiu, G., Andronie, M., Uţă, C., and Hurloiu, I. (2019). “Trust Management in Organic Agriculture: Sustainable Consumption Behavior, Environmentally Conscious Purchase Intention, and Healthy Food Choices,” Frontiers in Public Health 7: 340. doi: 10.3389/fpubh.2019.00340.

Pocol, C.B., Marinescu, V., Dabija, D.C., and Amuza, A. (2021). “Clustering Generation Z university students based on daily fruit and vegetable consumption: empirical research in an emerging market,” British Food Journal 123(8): 2705-2727. doi: 10.1108/BFJ-10-2020-0900.

Introduction

Please try to have a better emphasize and explanation of the

research gap

research question

theory which the paper is enhancing

As you are referring to Gen Z, your paper could, for instance, improve and enhance the generational theory. See for instance: https://doi.org/10.3390/su11174532

Lit review

After the lit review, please present the conceptual model under investigation

The lit review should be enhanced and extending. There should also be a paragraph on the theory that you employ

Methodology

Please present the money in EUR or USD

From the methodology it should be clear how the questionnaire has been operationalised, i.e. what scales have been used. There should be some analysis presented regarding reliability, validity etc. and the general procedure employed by the authors should also be presented.

Table 2. Please also state the items ... it is needed to know what the items are

Items below 0.7 must be dismissed

Please also present the cronbach alpha

Table 3. Is this the correlation matrix or the discriminant validity? This is not clear!

Table 4. for each hypothesis there should be a discussion

Discussions

Please make more comparisons between your findings and previous results from the literature, thus pinpointing the novelty of your research.

Conclusions

please also add limitations and future research perspectives

Please add more up to date references.

The paper needs proofreading

Reviewer #2: Title: Factors affecting repurchase intention of organic food among generation Z (Evidence from developing economy)

Comments: Major Revision

The current manuscript needs major improvement in terms of writing and APA formatting.

Write references in APA VI format.

Firstly, this manuscript needs to be prove-read by a native English Speaker and a research analyst before submitting it again.

My concern is that why the author has chosen to carry out research on such old and over used variables? I do not see any contribution of this current manuscript on the existing literature. In simpler words, what is new in this research?

Page 1 line no. 41, “It is noted that various studies have been conducted in western countries…” mention the research here.

Page 1 line 42, “Similarly, Biswas & Roy[3]…” cite properly

Write all statistics in one paragraph, consider Page 1, Para 1 & 2

Page 2 line 50, cite properly. Either cite through number or through names in order to maintain the coherence of the manuscript.

Page 2 line 54 “generation z” .Either write generation Z or generation z throughout the text.

The gap in the study is not addressed properly as most studies related to repurchase intention of organic food and generation Z are missing while developing a problem statement since these topics would strengthen your problem statement.

Refer to the following article while mentioning existing literature in the problem statement

https://doi.org/10.1108/MEQ-12-2021-0279

In the literature review section, the author has failed to mention any theory/s applicable to the research model, in detail at the beginning of the chapter. See page no. 3, heading “2. Theoretical framework and hypotheses”

Moreover, the manuscript lacks consistency as the author has not defined/describe each variable with respect 3-4 previous studies. Moreover, the author has not provided reference of previous literature with respect to the relationship between IV and DV, and created hypotheses just like that. Refer to Page no. 3-4. Similarly, the author has missed out some recent literature related to the variables.

For customer satisfaction, refer to:

• https://doi.org/10.1080/14783363.2015.1100517

• https://doi.org/10.1108/BFJ-11-2018-0728

For Brand awareness, refer to:

• https://doi.org/10.1108/IJOEM-08-2017-0275

• https://doi.org/10.1002/csr.1960

Page 4, line 141, the demographic cohort referred as Gen Z in this article ranges between the years 1996-2012 where as you have mentioned “1995-2010”. So, provide reference for using this range.

Referring to 4.1. Data Analysis and Results, Page 5 line no. 176. Have you applied SEM or PLS-SEM? Because all the tables you have shared are that of PLS-SEM not SEM neither CFA.

Where are the other tables of Discriminant validity (Cross-leadings and HTMT)?

The author must write detailed interpretation of all the tables along with providing the references for the benchmark values. Refer to the following articles:

https://doi.org/10.1108/TQM-02-2020-0019

https://doi.org/10.1177%2F0735633117715941

Referring to the “5. Discussion” page no. 6-7, first of all correct the numbering. It should be the part of chapter 4. Secondly, none of the result of the path analysis is discussed through the channel of existing literature. No reference is give, no proper justification is given with respect to Gen Z.

Where is the conclusion in chapter 5, author has just directly written “5.1.1 Theoretical Contribution”

Similarly referring to “5.1.2. Practical contribution”, the author has not given managerial implications properly as they are quite generalized and not linked to the results.

Write Future recommendation under separate heading. Hence, the author must rewrite chapter 5. See pages 7-8

6. PLOS authors have the option to publish the peer review history of their article (what does this mean?). If published, this will include your full peer review and any attached files.

Reviewer #1: No

Reviewer #2: No

---

## [Author Response · Author response to Decision Letter 0]

28 Nov 2022

i have incorporated all of required comments

---

## [Decision Letter · Decision Letter 1]

26 Jan 2023

factors affecting repurchase intention of organic food among generation Z (Evidence from developing economy)

PONE-D-22-18927R1

Dear Dr. Khan,

We’re pleased to inform you that your manuscript has been judged scientifically suitable for publication and will be formally accepted for publication once it meets all outstanding technical requirements.

Kind regards,

Ming Zhang, Ph.D.

Academic Editor

PLOS ONE

Additional Editor Comments (optional):

Reviewers' comments:

Reviewer's Responses to Questions

**Comments to the Author**

1. If the authors have adequately addressed your comments raised in a previous round of review and you feel that this manuscript is now acceptable for publication, you may indicate that here to bypass the “Comments to the Author” section, enter your conflict of interest statement in the “Confidential to Editor” section, and submit your "Accept" recommendation.

Reviewer #1: All comments have been addressed

2. Is the manuscript technically sound, and do the data support the conclusions?

Reviewer #1: Yes

3. Has the statistical analysis been performed appropriately and rigorously? 

Reviewer #1: Yes

4. Have the authors made all data underlying the findings in their manuscript fully available?

Reviewer #1: Yes

5. Is the manuscript presented in an intelligible fashion and written in standard English?

Reviewer #1: Yes

6. Review Comments to the Author

Reviewer #1: Thank you for implementing all my suggestions and recommendations. The paper can now be accepted. Best luck in attracting valuable citations

7. PLOS authors have the option to publish the peer review history of their article (what does this mean?). If published, this will include your full peer review and any attached files.

Reviewer #1: No

---

## [Editor Report · Acceptance letter]

30 Jan 2023

PONE-D-22-18927R1 

Factors affecting repurchase intention of organic food among generation Z (Evidence from developing economy) 

Dear Dr. Khan:

I'm pleased to inform you that your manuscript has been deemed suitable for publication in PLOS ONE. Congratulations! Your manuscript is now with our production department. 

Kind regards, 

on behalf of

Dr. Ming Zhang 

Academic Editor

PLOS ONE